# Associations between Milk Fatty Acid Profile and Body Condition Score, Ultrasound Hepatic Measurements and Blood Metabolites in Holstein Cows

**DOI:** 10.3390/ani12091202

**Published:** 2022-05-06

**Authors:** Diana Giannuzzi, Alessandro Toscano, Sara Pegolo, Luigi Gallo, Franco Tagliapietra, Marcello Mele, Andrea Minuti, Erminio Trevisi, Paolo Ajmone Marsan, Stefano Schiavon, Alessio Cecchinato

**Affiliations:** 1Department of Agronomy, Food, Natural Resources, Animals and Environment (DAFNAE), University of Padua, 35020 Legnaro, Italy; alessandro.toscano@unipd.it (A.T.); sara.pegolo@unipd.it (S.P.); luigi.gallo@unipd.it (L.G.); franco.tagliapietra@unipd.it (F.T.); stefano.schiavon@unipd.it (S.S.); alessio.cecchinato@unipd.it (A.C.); 2Department of Agricultural, Food and Agro-Environmental Sciences, University of Pisa, 56124 Pisa, Italy; marcello.mele@unipi.it; 3Department of Animal Science, Food and Nutrition (DIANA), The Romeo and Enrica Invernizzi Research Center for Sustainable Dairy Production (CREI), Faculty of Agricultural, Food and Environmental Sciences, Università Cattolica del Sacro Cuore, 29122 Piacenza, Italy; andrea.minuti@unicatt.it (A.M.); erminio.trevisi@unicatt.it (E.T.); paolo.ajmone@unicatt.it (P.A.M.); 4Nutrigenomics and Proteomics Research Center, Università Cattolica del Sacro Cuore, 29122 Piacenza, Italy

**Keywords:** blood metabolites, dairy cows, fatty acids, multivariate factor analysis, stress

## Abstract

**Simple Summary:**

Metabolic disorders represent a crucial problem in early lactating dairy cows, which lead to major economic losses at the herd level. To allow the prompt detection of metabolic dysfunction with noninvasive and ready-to-use methods on the farm, milk matrix represents the best option. Among milk fine components, the fatty acid profile represents a fingerprint of the cow’s nutritional and metabolic status, being a suitable indicator of metabolic imbalance at the cow level. We performed an association study between milk fatty acid profile and a set of metabolic indicators, such as body condition score, ultrasound liver measurements, and blood metabolites, from 297 Holstein–Friesian cows. We extracted a few latent variables able to explain specific biological mechanisms from the milk fatty acid profile. Then, we explored the associations between these new synthetic variables, namely the factors, and the morphometric, ultrasonographic and hematic indicators of immune and metabolic status. The significant associations of fatty acid factors with blood and ultrasound indicators of inflammation and hepatic load showed the capacity of fatty acids to reflect the energy metabolic status of lactating cows, suggesting their potential usefulness as markers of digestive alterations and metabolic variations in cows during the critical period of early lactation.

**Abstract:**

Dairy cows have high incidences of metabolic disturbances, which often lead to disease, having a subsequent significant impact on productivity and reproductive performance. As the milk fatty acid (FA) profile represents a fingerprint of the cow’s nutritional and metabolic status, it could be a suitable indicator of metabolic status at the cow level. In this study, we obtained milk FA profile and a set of metabolic indicators (body condition score, ultrasound liver measurements, and 29 hematochemical parameters) from 297 Holstein–Friesian cows. First, we applied a multivariate factor analysis to detect latent structure among the milk FAs. We then explored the associations between these new synthetic variables and the morphometric, ultrasonographic and hematic indicators of immune and metabolic status. Significant associations were exhibited by the odd-chain FAs, which were inversely associated with β-hydroxybutyrate and ceruloplasmin, and positively associated with glucose, albumin, and γ-glutamyl transferase. Short-chain FAs were inversely related to predicted triacylglycerol liver content. Rumen biohydrogenation intermediates were associated with glucose, cholesterol, and albumin. These results offer new insights into the potential use of milk FAs as indicators of variations in energy and nutritional metabolism in early lactating dairy cows.

## 1. Introduction

Metabolic disorders are a critical problem in dairy cows, and often underlie the onset of disease [1]. These dysfunctions mainly concern the failure of individual animals to cope with complex nutritional and metabolic processes and to adapt to large variations in them during early lactation [2]. Although poor adaptation may start before calving, often it happens without clinical signs [3,4]. Over the years, various indicators have been tested, alone or in combination, with the aim of discovering non- or minimally invasive techniques for the early detection of metabolic disorders at the herd level [5]. For example, body condition scores (BCS) that are excessively high at calving or that decrease sharply in early lactation have been associated with susceptibility to health disorders, such as milk fever, fatty liver, and ketosis [6]. Pronounced increases in the size and thickness of the liver and in portal vein blood flow measured by ultrasound (US) have been shown to give good indications of fatty liver conditions [7,8]. Liver triacylglycerol (TAG) content predicted by US texture analysis has also been identified as a promising tool to detect fatty liver [9]. Furthermore, thresholds of blood metabolites, such as non-esterified fatty acids (NEFA), β-hydroxybutyrate (BHB) [10], total bilirubin [11], aspartate aminotransferase (AST), gamma-glutamyl transferase (GGT), and haptoglobin [12], have been associated with ketosis and subsequent higher risk of postpartum diseases, poorer reproductive performance, and lower milk production [13,14]. To identify these parameters in the blood, the cow needs to be captured and a blood sample taken, which must be done by a veterinarian, with associated costs and time. The milk matrix, on the other hand, which can be regarded as a combination of compounds with different physiological meanings and hence a source of information for the prediction of animal health status, can be analyzed from milk samples collected during normal milking operations.

Bovine milk fat contains approximately 400 different fatty acids (FAs), which are derived almost equally from mammary gland synthesis and transfer from circulating plasma. They result from the mobilization of adipose body reserves and the processing of the diet during ruminal biohydrogenation, dietary digestion and intestinal absorption [15]. De novo synthesis in the mammary gland yields short- and medium-chain FAs (C4 to C14) and roughly half of the C16 FAs that originate from acetate and β-hydroxybutyrate [16]. The remaining C16 and all the longer chain FAs are derived either from dietary metabolism or fat depot mobilization [17,18]. In addition, odd- and branched chain FAs are generated by bacteria during ruminal activity.

The milk FA profile therefore represents a fingerprint of the cow’s nutritional and metabolic status and can vary substantially in accordance with changes in feeding regimes and management practices [19,20]. As such, milk FAs can be considered putative indicators of energy and metabolic status at the cow level.

To capture the complex relationships among FAs, researchers have found that analyzing simultaneous variations in groups of FAs is more efficient than analyzing variations in individual FAs [20,21,22]. Statistical approaches that use data reduction methods, such as multivariate factor analysis (MFA), can be applied to elucidate the network of correlations among FAs and to extract a few latent, phenotypically independent variables able to describe specific metabolic mechanisms [23,24].

In this study, we used the milk FA profile of 297 early lactating Holstein–Friesian cows as a potential source of information for metabolic status. Specifically, our aims were: (i) to extract latent variables from the milk FA profile to be used as novel and more informative phenotypes identifying the specific metabolic pathways of FA; (ii) to explore the associations between these new synthetic variables and various indicators of metabolic status, including BCS, direct and predicted liver US measurements, and a set of hematochemical parameters; and (iii) to identify functionally-related groups of FA as potential indicators of metabolic variations.

## 2. Materials and Methods

### 2.1. Animal Data, Milk and Blood Sampling

The present study is part of a broader project (BENELAT), involving more than 1000 lactating cows, aimed at developing short- and long-term interventions for improving animal welfare and efficiency, and the quality of dairy cattle production. The animals were reared in two herds located in Piacenza Province (northwestern Italy); they were housed in free stalls and fed total mixed ration (TMR). The ingredients and chemical compositions of the diets are reported in Table 1. Drinking water was provided by automatic water bowls. The project was approved by the ethical committee of the Organismo Preposto al Benessere degli Animali (OPBA; Organization responsible for animal welfare) of the Università Cattolica del Sacro Cuore and by the Italian Ministry of Health (protocol number 510/2019-PR of 19 July 2019). All procedures were performed in accordance with the relevant guidelines and regulations.

In the present study, we focused on cows in the first phase of lactation since they are more susceptible to metabolic stress. Individual milk samples were collected once during the evening milking from 297 Holstein–Friesian cows (herd A *n*= 262 and herd B *n*= 35) without signs of clinical disease and selected according to their days in milk (DIM; in the range 5–120 days postpartum). The milk production in the two herds was comparable (herd A: 36.89 ± 8.33 and herd B: 38.16 ± 7.29 kg/d).

Milk samples were divided into two aliquots: one was used for quality and composition analysis, which was performed by the Breeders Association of the Veneto, while the other was taken to the laboratory of the Department of Agronomy, Food, Natural Resources, Animals and Environment (DAFNAE) of the University of Padua (Legnaro, Padua, Italy) to determine the FA profile.

Blood samples (5 mL) were collected from the jugular vein in the morning before TMR distribution and placed in vacuum tubes containing 150 lithium heparin USP units (Vacumed; FL Medical, Torreglia, Padua, Italy). Blood and milk sampling, liver US and BCS were carried out on the same day for each cow from September 2019 to February 2020 (two or more sampling days per herd, depending on the farm size; 14 different herd/date combinations).

### 2.2. Indicators of Metabolic Stress

#### 2.2.1. Ultrasonographic Measurement Acquisition

Details regarding the acquisition of the US measurements are given in previous studies [5,9,25]. Briefly, a veterinarian with extensive experience in the liver US technique carried out transcutaneous US imaging of the liver at the 10th intercostal space on the right side of the standing animal. Images of the hepatic parenchyma were captured with a Mylab OneVET portable US scanner (Esaote SpA, Genoa, Italy) connected to a linear probe (Animal Science Probe, SV3L11; Esaote SpA, Genoa, Italy). The US frequency (2.8 MHz) and depth (21 cm) settings for visualizing the hepatic parenchyma were kept constant for all the animals enrolled in the study. The liver was also examined to exclude the presence of focal lesions not related to metabolic dysfunctions (i.e., abscesses, neoplastic masses). The final US images for each animal were selected by a single operator based on its diagnostic capacity. Liver depth (LD, mm), portal vein area (PVA, mm^2^), and portal vein depth (PVD, mm) were measured using the MyLab Desk software (Esaote SpA, Genoa, Italy). For prediction of the triacylglycerol (pTAG, mg/g) contents, the hepatic parenchyma was analyzed with the MaZda v.4.6 texture analysis software (Technical University of Lodz, Institute of Electronics, Lodz, Poland).

#### 2.2.2. Hematochemical Parameters Analysis

Blood metabolic profile was obtained in accordance with the protocols described by Mezzetti et al. [26]. Briefly, blood samples were kept on ice until centrifugation at 3500× *g* for 16 min at 6 °C (Hettich Universal 16R Centrifuge), which was performed within 2 h of collection. A small fraction of blood was used to determine hematocrit (packed cell volume) (ALC Centrifugette 4203; 15,300× *g*, 12 min). Briefly, blood is drawn into a capillary tube by capillary forces and centrifugated, and then the ratio of the column of packed erythrocytes to the total length of the sample in the capillary tube, measured with a graphic reading device, is calculated and expressed as a decimal.

The plasma thus obtained was stored at −20 °C until analysis. A clinical auto-analyzer (ILAB-650, Instrumentation Laboratory-Werfen, Bedford, MA, USA) was used to determine the concentrations of glucose, NEFA, BHB, urea, creatinine, calcium, phosphorus, magnesium, sodium, potassium, chlorine, zinc, AST, GGT, alkaline phosphatase (ALP), total protein, haptoglobin, ceruloplasmin, albumin, total bilirubin, cholesterol, globulin, total reactive oxygen metabolites (ROMt), advanced oxidation protein products (AOPP), ferric reducing antioxidant power (FRAP), total thiol groups (SHp), paraoxonase, and myeloperoxidase.

#### 2.2.3. Body Condition Score

The same operator assigned BCS to all the cows according to Edmondson et al. [27] on a scale ranging from 1 (very lean) to 5 (very fat) in increments of 0.25. This trait was analyzed as reported in Piazza et al. [28]. Briefly, we used a mixed model that included the fixed effects of DIM and parity, and the random effect of herd/date, allowing the classification of the cows into three classes according to their residuals from the mixed model (low [lean], high [fat] and medium [standard] for residuals of <−0.75, >0.75, and between −0.75 and 0.75 standard deviation units, respectively).

### 2.3. Milk Fatty Acid Analysis

#### 2.3.1. Sample Preparation and Lipid Extraction

All milk samples were freeze-dried, after which a variable quantity of lyophilized milk was weighed in order to obtain nearly 40 mg of lipids. Prior to analysis with gas chromatography, FA methyl esters were obtained by a direct transesterification procedure of milk samples, using a combined basic and acid procedure able to transesterify all the milk lipid classes: triglycerides, diglycerides, monoglycerides, phospholipids, sphingolipids, cholesterol esters, and free FAs, according to Christie [29]. Briefly, the samples were heated in two successive water baths (15 min each): the first at 50 °C with 1 mL of sodium methoxide (0.5 M in methanol), the second at 80 °C with 1.5 mL of 5% methanolic HCl, with each followed by a cooling stage. Finally, after adding 2 mL of hexane and 2.5 mL of 6% K_2_CO_3_ the solution was centrifuged, and the supernatant was collected for gas chromatography.

#### 2.3.2. Gas Chromatography

Fatty acid methyl esters (FAME) were analyzed using a 7820A GC System (Agilent Technologies, Santa Clara, CA, USA) fitted with an automatic sampler (Agilent 7693, Santa Clara, CA, USA) and a flame ionization detector (FID) connected to chromatography data system software (OpenLab CDS, Agilent ChemStation, Santa Clara, CA, USA). Separations were performed on SLB-IL 111 columns (100 m × 0.25 mm, 0.2 µm film thickness; Merck KGaA, Darmstadt, Germany), with a constant flow of hydrogen as carrier gas (0.6 mL/min). The temperature gradient was as follows: 40 °C for 4 min, then 50 °C/min to 150 °C; after 30 min 2 °C/min up to 200 °C and maintained for 30 min. Finally, the temperature was raised to 240 °C (ΔT= 4 °C/min) and maintained until the end of the analysis (75 min in total). The FID operated at 250 °C, and the injection port at 300 °C, with a split ratio of 1:80 (1 µL of sample injected). Forty-three FA peaks were identified by comparison with the peaks generated by injection of Supelco 37 Component FAME Mix (Sigma-Aldrich and Nu-Chek Prep Inc., Elysian, MN, USA) and Supelco conjugated (9Z,11E)-linoleic acid (Sigma-Aldrich). The remaining unidentified peaks were labelled using the analytical methods proposed by [30].

### 2.4. Statistical Analysis

#### 2.4.1. Multivariate Factor Analysis

Multivariate factor analysis assumes that the variance in each original variable can be decomposed into its common and unique components, termed communality and uniqueness, respectively. In matrix notation, the factor model decomposes the covariance matrix of the measured traits (S) as follows:S = BB′ + Ψ,(1)
where BB′ refers to the communality and Ψ the uniqueness (co)variance matrices [31]. According to the (co)variance model, the measured traits can be represented as a combination of p unobservable common factors (X) plus a residual specific variable (e) [31]:$$y_{1}=b_{11}X_{1}+\cdots +b_{1p}X_{p}+e_{1}$$
(2)$$y_{n}=b_{n1}X_{1}\cdots +b_{\mathrm{np}}{\;X}_{p}+e_{n},$$
where b*_ij_* are factor coefficients (or loadings, correlations between the *j*th common factor), and e*_i_* is the *i*th residual specific variable [31]. Loadings are the elements of the B matrix used in the factor model. Common factors generate covariances among measured variables, whereas the residual contributes only to the individual variation.

All statistical analyses were performed with the R software v.3.6.3 (www.r-project.org, accessed on 1 December 2021). The factor analysis was carried out with the *psych* R package on the correlation matrix of 46 variables, i.e., 43 relevant individual FAs according to Mele et al. [20], and three milk production traits (milk yield, fat and protein, kg/d), measured on the 297 cows. To assess the adequacy of the sampled data for MFA and quantify the difference between the Pearson and partial correlations of the measured variables, the Kaiser–Meyer–Olkin (KMO) measure of sampling adequacy (MSA) was calculated, a high MSA revealing the presence of a latent structure in the data [32]. *Varimax* factor rotation with the *minres* factoring method was then used to detect the very simple structure. The number of factors extracted was based on their eigenvalue (>1), their readability in terms of the relationships with the original variables, and the amount of explained variance [20]. Only variables exhibiting moderate/high correlations >|0.60| were considered associated with the factor [23].

#### 2.4.2. Mixed Model Analysis

We adopted a conservative approach to studying the associations between the latent variables and metabolic (MET) indicators and did not assume any linear relationship between the response and the independent variables. To correctly interpret the variations among the MET indicators (i.e., pTAG, PVD, LD, PVA and the 29 hematochemical parameters), continuous variables were discretized and classes were created on the basis of the 25th and 75th percentiles, with the exception of BCS, which was divided into three classes as previously described.

First, to explore the effects of individual sources of variation in the latent factors, we applied the following mixed linear model in the *lme4* R package:(3)$$y_{ijkl}=\mu +DIM_{i}+Parity_{j}+Herd/Date_{k}+e_{ijklm}$$
where *y**_ijkl_* is the observed trait (i.e., the latent factor scores); *µ* is the overall mean; *DIM_i_* is the fixed effect of the *i*th class of days in milk (*i* = 4 classes; class 1 ≤ 30 *n* = 39; 30 < class 2 ≤ 60 *n* = 71; 60 < class 3 ≤ 90 *n* = 77; 90 < class 4 ≤ 120 *n* = 110); *parity_j_* is the fixed effect of the *j*th parity (*j* = primiparous *n* = 195, multiparous *n* = 102); *Herd*/*Date_l_* is the random effect of the *l*th herd/date (*l* = 1 to 14); e*_ijklm_* is the random residual.

We then used a model with an increasing complexity with respect to model 3 to assess the associations between the latent factors and the MET indicators; the latter were introduced into the model 4 one at a time to avoid potential multicollinearity problems between the 34 predictors:(4)$$y_{ijklm}=\mu +DIM_{i}+Parity_{j}+MET\_ind_{k}+Herd/Date_{l}+e_{ijklm\;}$$
where *y_ijklm_* is the observed trait (i.e., the latent factor scores); *µ* is the overall mean; *DIM_i_* is the fixed effect of the *i*th class of days in milk (*i* = 4 classes; class 1 ≤ 30; 30 < class 2 ≤ 60; 60 < class 3 ≤ 90; 90 < class 4 ≤ 120); *parity_j_* is the fixed effect of the *j*th parity (*j* = primiparous, multiparous); *MET_ind_k_* is the fixed effect of the *k*th class of the metabolic indicators discretized on the basis of the 25th and 75th percentiles (or the class of BCS), as previously described; *Herd/Date_l_* is the random effect of the *l*th herd/date (*l* = 1 to 14); *e**_ijklm_* is the random residual. Herd/Date and the residuals were assumed to be normally distributed with a mean of zero and variances of $\sigma_{h}^{2}$ and $\sigma_{e}^{2}$, respectively. The restricted maximum likelihood method was used to estimate the variance components. The proportion of variance explained by herd/date was calculated by dividing the corresponding variance component by the total variance. A given effect was considered significant at *p* < 0.05. Polynomial contrasts (*p* < 0.05) were used to determine the response curve of the data for the MET indicators; first-order comparisons measured the linear relationships, while second order comparisons measured quadratic relationships. Only significant results are displayed in the figures.

## 3. Results

### 3.1. Metabolic Indicators and Latent Factors of Milk Fatty Acids

The set of milk FA analyzed comprised 43 individual FAs: 22 saturated, 12 monounsaturated, and 9 polyunsaturated. Detailed descriptive statistics of the production traits and the individual FAs are reported in Appendix A, while descriptive statistics for all indicators of metabolic stress (i.e., BCS, US measurements, and hematochemical parameters) are given in Appendix A. The FA dataset had an MSA value of 0.75, revealing good suitability for MFA [33]. The MFA extracted eight latent factors (F) explaining 68% of the total variance. Only 37 of the 43 FAs contributed to the generation of the latent factors. As shown in Table 2, each variable was highly correlated with only one factor and weakly with all the others, confirming the adequacy of factor model assumptions for the data.

### 3.2. Individual Sources of Variation

Results (*p*-values and RMSE) from the linear mixed model used to estimate the individual sources of variation (i.e., DIM, parity) that affect the latent factors are reported in the first two rows of Table 3 and Table 4.

Days in milk affected F1-de novo synthesis/preformed, with factor loadings increasing linearly at increasing DIM (*p* < 0.001). Order of parity affected the production factor, the loadings for multiparous being higher than for primiparous (*p* = 0.01).

The herd/date random effect, which also includes feeding regime, explained between 1 and 13% of the total variation for all the latent factors except F2- branched chain fatty acids (BCFA), F4-alternative rumen biohydrogenation (RBH) ALA, and F8-RBH, which exhibited herd/date variances of around 17, 25 and 40%, respectively (data not shown in table).

### 3.3. Associations between Metabolic Indicators and Factors

Results of the associations between the set of metabolic indicators and F1 and F2 are reported in Table 3 and Figure 1. Increasing AOPP values and ceruloplasmin plasma concentrations (Figure 1a,b; *p* = 0.014 and *p* = 0.010 respectively) were associated with decreasing scores in F1-de novo synthesis/preformed factor. Increasing liver pTAG content (Figure 1c; *p* = 0.028) and decreasing of cholesterol *(p* = 0.046), AOPP (*p* = 0.017) and ceruloplasmin (*p* = 0.030) serum levels (Figure 1d–f, and respectively) were associated with decreased F2-BCFA scores.

Significant associations between metabolic indicators and F3 and F4 are reported in Table 3 and Figure 2. Lower BCS was associated with lower scores for the factor that included long chain—polyunsaturated fatty acids (LC-PUFA; Figure 2a; *p* = 0.007). On the other hand, in the case of ALP and SHp, the associations tended to follow a quadratic pattern, with higher scores for F3-LC-PUFA moving from medium to high levels than from low to medium plasma levels (Figure 2b,c; *p* = 0.015 and *p* = 0.044 respectively). Finally, increasing plasma concentrations of ceruloplasmin were associated with linear increases in F3-LC-PUFA (Figure 2d; *p* = 0.012). Decreasing liver pTAG contents and NEFA plasma concentrations were associated with linear decreases in F4-alternative RBH ALA scores (Figure 2e,f; *p* = 0.025 and *p* = 0.015). Whereas, increasing glucose and ALP were associated with linear increases in F4-alternative RBH ALA scores (Figure 2g,h; *p* = 0.019 and *p* < 0.001).

Table 4 shows the associations between metabolic indicators and F5-production and F6-short-chain fatty acids (SCFA). Specifically, with increasing liver pTAG content, F5 scores decreased linearly (Figure 3a; *p* = 0.030) but with increasing cholesterol and albumin, plasma levels increased linearly (Figure 3b,c; *p* = 0.004 and *p* = 0.037, respectively). The short-chain fatty acid factor was inversely associated with liver pTAG content, with scores decreasing linearly with the increase in pTAG, although the decrease in scores tended to be greater moving from medium to high liver pTAG contents than from low to medium (Figure 3d; *p* = 0.044).

The F7-Odd-chain fatty acids (OCFA) exhibited the highest number of significant associations with metabolic indicators, all of which are reported in Table 4. Increasing GGT and SHp plasma concentrations appeared to be linearly and positively related to increasing F7 scores (Figure 4a,b; *p* = 0.046 and *p* = 0.032, respectively). The same positive linear pattern of association was observed between albumin plasma levels and F7 (Figure 4c; *p* < 0.001). The association between glucose and F7 exhibited a positive quadratic pattern, with greater increases in F7 scores moving from low to medium than from medium to high glucose plasma concentrations (Figure 4d; *p* = 0.039). In contrast, inverse associations were found between BHB and circulating proteins and F7 scores. Specifically, with increasing plasma concentrations of BHB, ceruloplasmin and globulins F7 decreased with a linear trend (Figure 4e–g; *p* = 0.003, *p* = 0.016, and *p* = 0.021, respectively); in contrast, total proteins exhibited a quadratic pattern, with medium plasma levels showing higher F7 scores compared to low and high plasma levels (Figure 4h, *p* = 0.023).

The associations between metabolic indicators and F8-RBH are reported in Table 4. Increasing energy-related blood metabolites, such as glucose and cholesterol were linearly and positively associated with F8 scores (Figure 5a,b; *p* = 0.016 respectively). The same pattern of association was observed for albumin and AOPP, with the highest albumin and AOPP values associated with the highest F8 scores (Figure 5c,d; *p* = 0.011 and *p* = 0.015 respectively).

## 4. Discussion

### 4.1. Latent Factors Identify Clusters of Fatty Acids Acting in Common Metabolic Pathways

In this study, we performed an MFA on the milk FA profile of Holstein–Friesian dairy cows in early lactation. The correlations between the original variables (i.e., individual FAs) and each latent factor were fairly straightforward. The first factor was named de novo synthesis/preformed and was positively correlated with medium-chain saturated FAs (C8 to C14), 10:1 c9, and 14:1 c9, and negatively correlated with 17:0, 17:0 iso, 17:0 ante, 18:0, 17:1 c9, 16:1 t9, and 18:1 c9. This factor explained more of the variability in our dataset than the other factors. In previous studies performing MFA of milk fatty acids [20,21,22], the metabolic pathway of de novo synthesis was represented by an independent latent variable. The unification into a single latent factor of two categories of FA with such diverse origins is probably related to the lactation stage of the cows in the present study, which are all in their first 120 DIM. In fact, mammary gland endogenous lipogenesis decreases when cows are rapidly mobilizing body fat, as occurs in early lactation [34]. In this phase, the two metabolic pathways of mammary gland endogenous FA synthesis and FA recovery from body reserves are the main mechanisms for meeting milk fat requirements.

The second factor is positively correlated with 13:0 iso, 14:0 iso, 15:0 iso, 15:0 ante, and 16:0 iso, thus identifying BCFA. Branched chain fatty acids originate from bacterial cellulolytic activity in the rumen, and they are found in greater amounts in milk from cows fed high fiber diets [35,36].

Long-chain fatty acids, especially long chain—polyunsaturated fatty acids (LC-PUFA), clustered in the third factor. This latent variable also includes α-linolenic acid (ALA), linoleic acid (LA) and cis 9, trans 11 conjugated LA (rumenic acid). LC-PUFA are present in milk fat in low amounts, deriving mainly from the diet and to a lesser extent from endogenous elongation of ALA and LA of dietary origin in the mammary gland [37,38].

The fourth factor was highly correlated only with the co-eluted MUFA t16 + c14 18:1, and moderately correlated with stearic acid (*r* = 0.43). This finding is unprecedented in similar studies. A range of 18:1 positional isomers are produced as intermediates during RBH of dietary LA and ALA to stearic acid and are then absorbed in the small intestine and incorporated into the milk fat [16]. According to Shingfield et al. [39], t16 18:1 in rumen liquor originates from RBH of ALA, an alternative pathway to that considered the major RBH; a shift in RBH is usually due to perturbing ingredients in the diet or to specific dietary regimes [39] and leads to a large number of *cis* and *trans* isomers of 18:1 and 18:2 being present in the milk fat. Unfortunately, specific silver ion chromatographic techniques are needed to identify all the *cis* and *trans* isomers in milk fat, and with the chromatographic technique adopted in the present work we were only able to determine t16 18:1 coeluted with c14 18:1, limiting our potential speculations on this factor.

The fifth factor identified the production traits, namely milk yield and milk composition (i.e., fat and protein). These relationships were very much expected due to the high correlations among these traits.

Short-chain saturated FAs (4:0, 6:0 and 8:0) were identified by the sixth factor. These FAs are endogenously synthesized in the mammary gland by acetyl-CoA carboxylase and fatty acid synthase enzymes.

Odd-chain saturated FAs 11:0, 13:0 and 15:0 were associated with the seventh factor, hence designated F7-OCFA. These FAs originate from ruminal metabolism and are synthesized from propionate through repeated condensation of malonyl-coenzyme A by ruminal microbial flora; a small number of them are also synthesized in the mammary gland [40].

The eighth factor grouped FAs originating from the RBH process. Although LA (18:2 c9, c12) did not exhibit high loadings on this factor and was instead associated with F1, the eighth factor was positively correlated with vaccenic acid (18:1 t11), the main product of LA RBH, and some other cis and trans isomers of 18:1, such as 18:1 c12, 18:1 t6–8, 18:1 t9, 18:1 t10, which are the end products of RBH of dietary FAs to stearic acid [41]. These intermediaries also play a role in the regulation of mammary lipogenesis, and therefore modify the milk FA profile both quantitatively and qualitatively.

### 4.2. Associations of Metabolic Indicators with Fatty Acid Latent Factors Reveal Their Role as Potential Markers of Metabolic Variations

We performed, for the first time, an association study between body and US measurements and a set of blood parameters and milk FA latent factors, to explore the relationships among metabolic variations and milk FAs in Holstein cows. A summary of the observed associations and their suggested physiological meanings is reported in Table 5. Briefly, SCFAs, BCFAs, and RBH ALA were positively associated with energy metabolism with an inverse relation with indicators of negative energy balance. De novo/preformed FAs and OCFAs exhibited positive relations with antioxidant activity and inverse association with inflammatory indicators and oxidation metabolites. Notably, OCFAs were inversely associated with BHB. On the other hand, LC-PUFAs showed a positive association with inflammatory status.

### 4.3. Indicators of Energy Metabolism

#### 4.3.1. Cholesterol

High concentrations of cholesterol are associated with high loadings on F2-BCFA, F5-production, and F8-RBH. Fermentation activity in the rumen in dairy cows influences not only the amount of BCFA in milk, as already discussed, but also total cholesterol synthesis from acetate [35,42]. In fact, the diet of dairy cows is very low in cholesterol, which has to be almost totally synthesized by the animal using acetate and glucose [42]. It could be argued that, under the same dietary conditions, more efficient ruminal fermentation produces higher amounts of acetate, and, consequently, cholesterol. Moreover, a higher level of cholesterol is favorable in early lactation, suggesting that the liver can easily synthesize lipoproteins [14] and consequently improve hepatic TAG export [43]. Therefore, individuals with higher serum cholesterol and lipoprotein concentrations are probably more efficient in controlling negative energy imbalance (NEB) as they are less prone to developing metabolic disorders and consequently more productive during early lactation [44]. Finally, the FA intermediates of RBH exhibited a positive linear relationship with cholesterol concentrations. Again, under favorable physiological conditions, more efficient rumen fermentation results in larger amounts of RBH byproducts being released into the bloodstream. Although there are no reports on the meaning of the associations between serum cholesterol and milk RBH byproducts, it is nonetheless the case that the latter are transferred to the milk from the bloodstream, so we can speculate that the intermediates of RBH in the milk reflect their analogues in the blood and their metabolic effects.

#### 4.3.2. Glucose, NEFA, and BHB

An increase in circulating NEFA was negatively associated with F4-alternative RBH ALA, while an increase in BHB serum levels was negatively associated with F7-OCFA. Serum glucose concentrations showed the opposite trend, with increases associated with increased loadings on both F4 and F7. As expected, circulating NEFA and BHB behave in opposite ways to glucose, as they represent the two components of the energy metabolism balance. During early lactation, high-yielding cows have to cope with extreme energy demands for milk production [45]. In this phase, glucogenic precursors, particularly ruminal propionate, which is massively converted into glucose in the liver, are often insufficient to meet immediate glucose needs, predisposing the cows to metabolic diseases and poor reproductive performance as a consequence of NEB [46]. Insufficient blood glucose triggers circulating NEFA increase. Nevertheless, the scarcity of energy (i.e., oxaloacetate) in the liver can impair NEFA oxidation, producing ketone bodies, such as BHB [45]. In this case, as our findings show, milk linear OCFAs (e.g., 13:0 and 15:0), which are de novo synthesized from propionyl-CoA by rumen bacteria or endogenously in the mammary gland, might provide information on the cow’s glucose status through their positive association with glucose, and on ketogenesis in the liver, through their negative association with BHB.

#### 4.3.3. pTAG

Overall, pTAG showed consistent trends: increasing levels of liver pTAG content were associated with linear decreases in F2-BCFA, F4-alternative RBH ALA, F5-production, and F6-SCFA. In our dataset, the third class of pTAG (>75 mg/g) comprises individuals with moderate—and therefore subclinical—fatty liver syndrome, a condition characterizing up to 65% of dairy cows during early lactation [47]. The accumulation of triacylglycerol in the liver results from a state of NEB, and when this condition is associated with other health problems, including severe inflammation [48], it progresses to its most severe forms with clinical manifestation, and reductions in milk production and feed intake [47]. Here, high values of liver pTAG content are associated with low loadings on the production factor, so that production traits exhibit a descending pattern when hepatic pTAG content rises. In addition, cows with higher levels of liver pTAG are less able to cope with NEB. In lactating cows, 80% of the glucose available in the organism is used by the udder [49]. At the same time, the synthesis of FA in the rumen (i.e., BCFA and isomers of 18:1), as well as in the mammary gland (i.e., SCFA), is dependent on the availability of glucose. Consistent with this, our findings show higher liver pTAG content to be associated with reduced amounts of SCFA, BCFA, and FA intermediates resulting from RBH grouped in F4.

#### 4.3.4. BCS

Analysis of BCS showed that the fatter cows in early lactation had higher loadings for LC-PUFA than leaner cows, confirming that, under the same dietary conditions, the milk FA profile, especially long-chain FAs, reflect changes in energy balance [50]. Indeed, LC-PUFAs derive almost entirely from the diet and high levels of them in the organism cause lipid accumulation [51], which explains the positive association with fatter cows in our populations.

### 4.4. Blood Metabolites Related to Inflammation

#### 4.4.1. Ceruloplasmin

We observed a range of associations between ceruloplasmin concentrations and FA latent factors: F1-de novo synthesis/preformed and F7-OCFA were negatively related to ceruloplasmin plasma levels, whereas F2-BCFA and F3-LC-PUFA were positively related. Ceruloplasmin is a positive acute phase protein (APP) that increases its serum concentration in response to inflammatory cytokines and is considered a promising marker of inflammation in dairy cows [14]. After calving, cows experience a systemic inflammatory state that inhibits feed intake and induces NEB [52]. If this paraphysiological condition is not quickly resolved during the first week of lactation, latent inflammation persists with suboptimal feed intake and circulation of pro-inflammatory cytokines (i.e., TNFα, IL1b, IL6), which have a detrimental effects on the cow’s health and productivity [53]. In clinically healthy individuals, higher levels of ceruloplasmin are suggestive of a reduced ability to maintain homeostasis. Furthermore, high dietary intake or circulating levels of OCFA have been associated with reductions in inflammatory conditions in several human diseases [54], with recent reports revealing that OCFA are also negatively associated with increases in APPs [55]. On the other side, ceruloplasmin is positively associated with F3-LC-PUFA, which comprised both the essential FA ALA and LA, respectively, the omega-3 and omega-6 FA precursors in mammals. This finding highlights the close connection between ceruloplasmin and inflammatory regulation. In addition, ceruloplasmin is also the major copper-carrying protein, and copper plays a crucial role in lipid metabolism, with respect to variations in cell membranes, in which ALA and LA are highly involved [56,57].

Regarding the positive association between ceruloplasmin and F2-BCFA, recent studies have shown that BCFA can elicit both an anti- and a proinflammatory response in various tissues [58], suggesting that these FAs could also affect (and reduce) the host’s immune response.

#### 4.4.2. Globulin and Protein Serum Levels

Globulin and total protein serum levels were negatively associated with factor 7-OCFA loadings, reflecting those observed for ceruloplasmin concentrations. It has been suggested that the concentration of serum globulins, which includes positive APPs, could be an indicator of the animal’s immune response [59,60]. Accordingly, the amount of OCFA was positively associated with albumin, a negative APP that decreases at the onset of inflammation [61] and is found in low concentrations for several weeks after a severe inflammation response in early lactating cows [62]. Albumin was also positively associated with production traits, as previously reported [44,59], and, as the major carrier of LCFA, including LA and ALA, with biohydrogenation activity in the mammalian organism [63].

### 4.5. Hepatic Enzymes and Antioxidants

#### 4.5.1. GGT

Increasing concentrations of the hepatic enzyme GGT were associated with an increase in the F7-OCFA factor loadings. As previously discussed, the amount of OCFA in milk is mainly related to ruminal microorganism synthesis and partly to endogenous production in the mammary gland. On the other hand, GGT is one of the most reliable markers of liver damage and oxidative stress and is susceptible to variation with increased liver activity during early lactation [64,65]. Vlaeminck et al. [66] observed that the milk OCFA profile differed from the intestinal OCFA profile and was more closely related to the profile in plasma TAG. As the amount of circulating TAG depends on the efficiency of liver TAG export, i.e., the capacity of the liver to package TAG into very-low-density lipoproteins [67], we might speculate that during the intense lipomobilization in the first phases of lactation high levels of GGT and TAG are released into the blood stream, and at the same time higher levels of OCFA can be found in the milk.

#### 4.5.2. ALP

Increased concentrations of ALP had a positive quadratic association with the F3-LC-PUFA group. ALP is correlated with APPs in metabolic syndrome [68], as it is a marker not only of liver damage but also of inflammatory conditions. It is therefore unsurprising to find a positive association with F3-LC-PUFA, which includes the main precursor of inflammation mediators. The same conclusions may be drawn regarding the positive association between ALP and the alternative RBH ALA factor, i.e., that it is due to the proinflammatory effects of the RBH byproducts [69].

#### 4.5.3. AOPP and SHp

Various FA factors were associated with AOPP. AOPP detection is a means of estimating the degree of oxidant-mediated protein damage [70]. This marker is inversely related to F1-de novo synthesis/preformed FA and positively related to F2-BCFA and F8-RBH. In agreement with our findings, Li et al. [71] showed that cell oxidative stress damage may be closely related to a decrease in endogenous milk fat synthesis by the mammary gland. Conversely, BCFAs are saturated FAs that are directed towards oxidative pathways [66], as are RBH byproducts, which are generated during the conversion of PUFAs into saturated FAs, and also activate the production of oxidative metabolites. We can therefore speculate that the FAs comprised in F2 and F8 are involved in the oxidative process of proteins and indirectly in their end products (i.e., AOPP).

Total thiol groups, as markers of the cow’s antioxidant status, were instead positively associated with F3-LC-PUFA and F7-OCFA. In agreement with the negative association between ceruloplasmin and OCFA, recent studies have pointed to the antioxidant and anti-inflammatory features of OCFA [54]. Regarding the F3-LC-PUFA in our dataset, it included omega-3 ALA and rumenic acid, which are well known to enhance antioxidant defenses [72,73]. These results seem in contrast to our findings regarding the association between ceruloplasmin, ALP, and F3-LC-PUFA, the latter seeming to enhance the proinflammatory response (and consequently oxidative stress). It is indeed the case that LC-PUFAs are precursors of various mediators that can elicit opposing responses when faced with different stress conditions. Further investigations drawing on a larger sample size that also includes cows affected by metabolic disorders are needed to clarify whether milk LC-PUFAs can be separated into different latent factors depending on their main inflammatory functions.

## 5. Conclusions

Our study confirms MFA as an effective statistical tool for reducing the complexity of the FA system while maintaining the biological significance and behavior of the original variables. Indeed, MFA grouped together FAs with similar origins and functions, reflecting common or related metabolic pathways and helping interpretation of all the associations with metabolic indicators.

We observed several associations between milk FAs and indicators of metabolic variations in dairy cattle. Of particular note, energy metabolites, hepatic enzymes (i.e., GGT and ALP), and blood indicators of inflammation (i.e., ceruloplasmin and globulins) were associated with F7-OCFA. These results could be exploited during farm-level screening to assess the welfare status in early lactating cows, to identify the precocious stage of inflammation or hepatic overload. Notably, OCFAs were inversely associated with ketonemia, underpinning their potential use as indicator of negative energy balance. This will only be possible along with employment of a rapid, non-invasive, easy-to-use tool, like infrared spectroscopy, able to predict FA scores at the population level, which will be a matter for further investigation. In addition, further studies also including animals with clinical disease are needed to validate the associations between metabolic variations during the critical first phase of lactation and changes in the milk FA profile.

## Figures and Tables

**Figure 1 animals-12-01202-f001:**
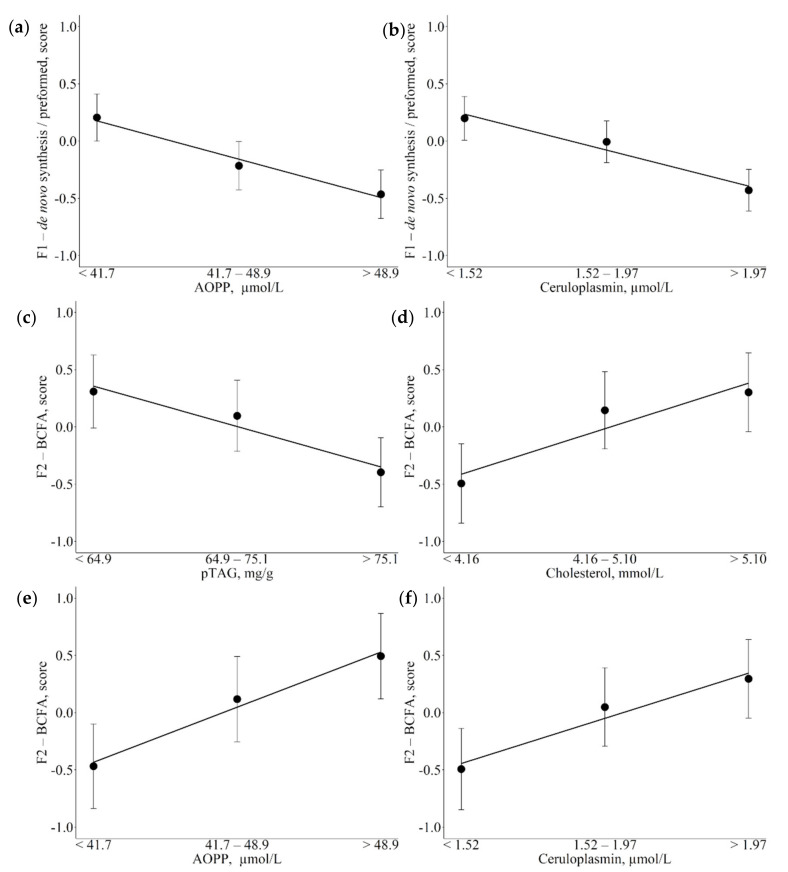
Least square means (LSM) for F1-de novo synthesis/preformed (**a**,**b**) and F2-branched chain fatty acids (BCFA; (**c**–**f**)) across the traits of concern. (**a,e**): advanced oxidation protein products (AOPP); (**b**,**f**): ceruloplasmin; c: predicted liver triacylglycerol (pTAG); (**d**): cholesterol. Black dots indicate the LSM and error bars indicate the standard error. Trendlines define linear or quadratic pattern according to polynomial contrast calculations (*p* < 0.05).

**Figure 2 animals-12-01202-f002:**
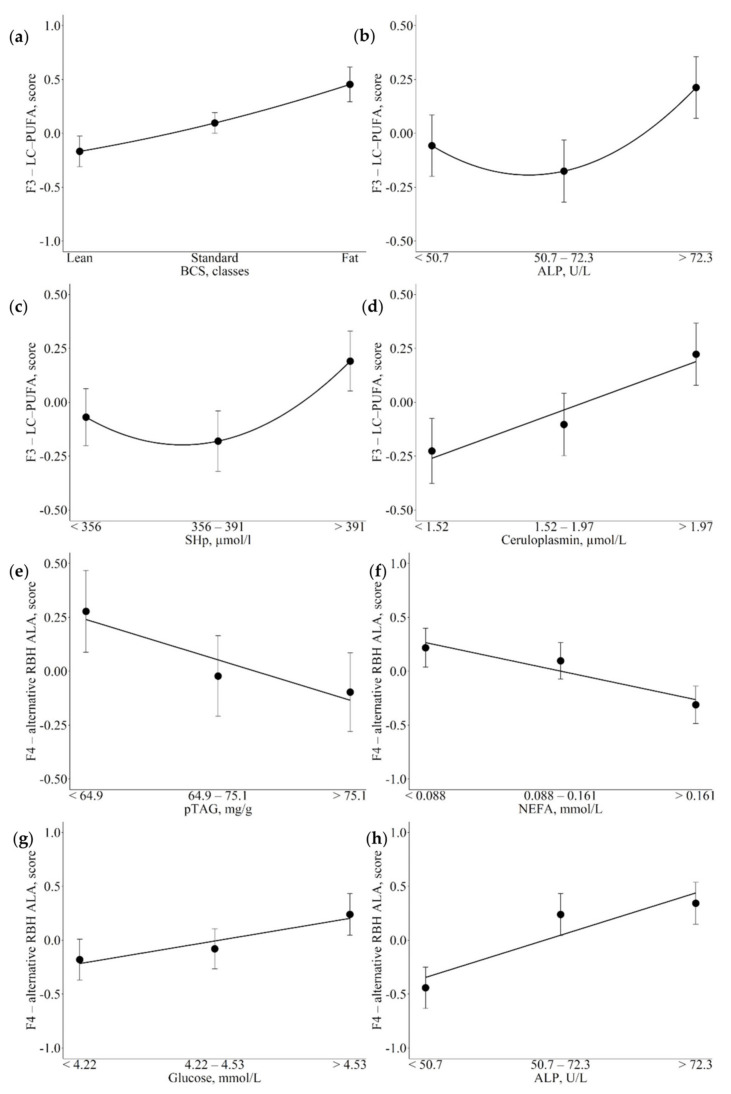
Least square means (LSM) for F3-long chain—polyunsaturated fatty acids (LC-PUFA; (**a**–**d**) and F4-alternative rumen biohydrogenation of α-linolenic acid (alternative RBH ALA) (**e**–**h**) across the traits of concern. (**a**): body condition score (BCS); (**b**,**h**): alkaline phosphatase (ALP); (**c**): total thiol groups (SHp); (**d**): ceruloplasmin; (**e**): predicted liver triacylglycerol (pTAG); (**f**): non-esterified fatty acids (NEFA); (**g**): glucose. Black dots indicate the LSM and error bars indicate the standard error. Trendlines define linear or quadratic pattern according to polynomial contrast calculations (*p* < 0.05).

**Figure 3 animals-12-01202-f003:**
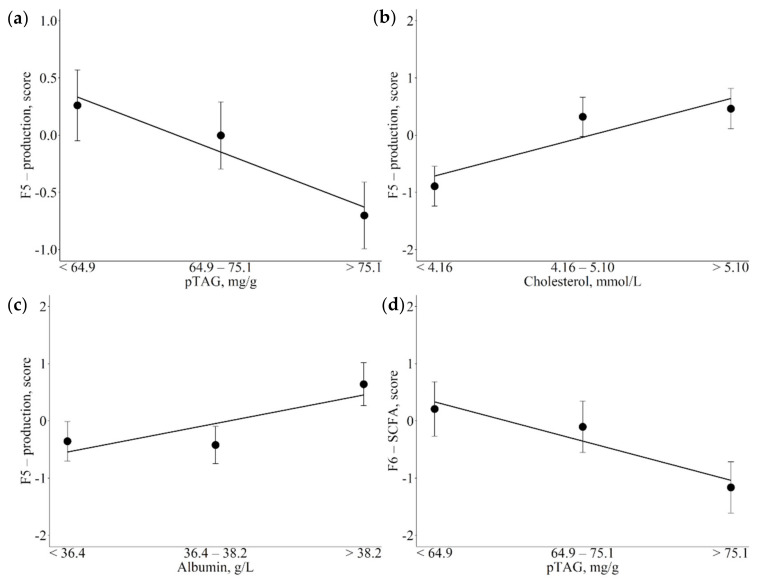
Least square means (LSM) for F5-production (**a**,**b**,**c**) and F6-short chain fatty acids (SCFA; (**d**)) across the traits of concern. (**a**,**d**): predicted liver triacylglycerol (pTAG); (**b**): cholesterol; (**c**): albumin. Black dots indicate the LSM and error bars indicate the standard error. Trendlines define linear or quadratic pattern according to polynomial contrast calculations (*p* < 0.05).

**Figure 4 animals-12-01202-f004:**
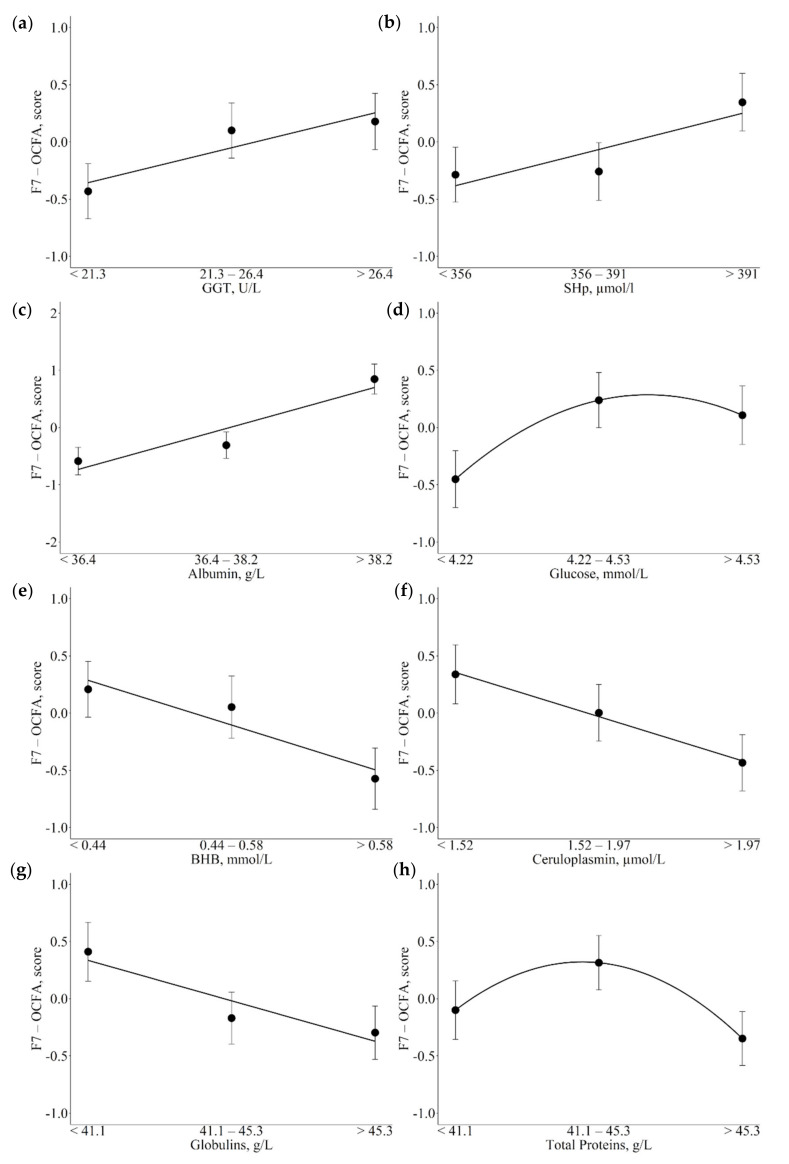
Least square means (LSM) for F7-odd chain fatty acids (OCFA) across the traits of concern. (**a**): = γ-glutamyl transferase (GGT); (**b**): total thiol groups (SHp); (**c**): albumin; (**d**): glucose; (**e**): β-hydroxybutyrate acid (BHB); (**f**): ceruloplasmin; (**g**): globulins; (**h**): total proteins. Black dots indicate the LSM and error bars indicate the standard error. Trendlines define linear or quadratic pattern according to polynomial contrast calculations (*p* < 0.05).

**Figure 5 animals-12-01202-f005:**
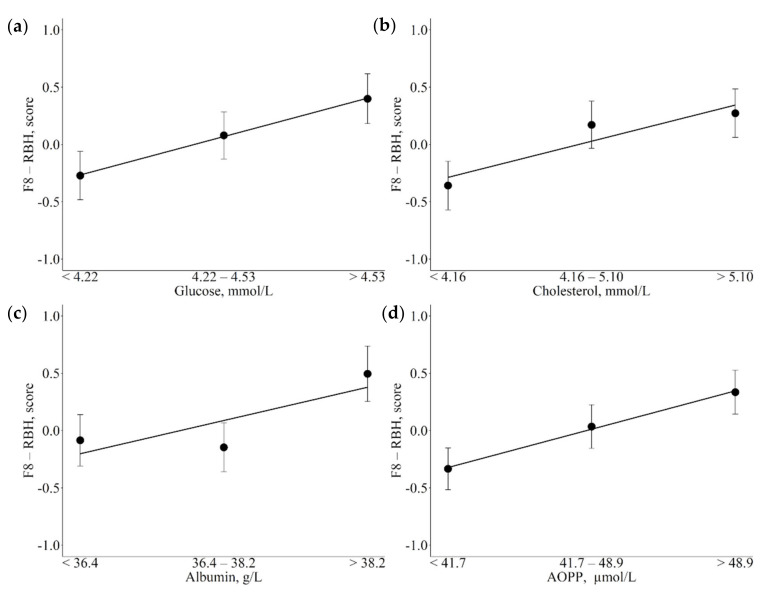
Least square means (LSM) for F8-rumen biohydrogenation (RBH) across the traits of concern. (**a**): glucose; (**b**): cholesterol; (**c**): albumin; (**d**): advanced oxidation protein products (AOPP). Black dots indicate the LSM and error bars indicate the standard error. Trendlines define linear or quadratic pattern according to polynomial contrast calculations (*p* < 0.05).

**Table 1 animals-12-01202-t001:** Diet composition (% of DM) during lactation and the dry period in the two herds.

	Herd A ^1^	Herd B ^1^	
	Lactation	Lactation Primiparous	Lactation Multiparous
Corn silage	12.42	30.76	32.94
Sorghum silage	24.6	-	-
Wheat silage	-	9.49	9.49
Alfalfa hay	20.98	11.47	11.48
Ryegrass hay	2.12	-	-
Barley straw	-	-	-
Corn grain ground	12.62	23.95	21.16
Barley grain ground	8.41	-	-
Soybean meal (solvent ex. 44% CP)	12.1	9.25	10.65
Sunflower meal (solvent ex. 25% CP)	4.34	4.69	4.43
Soybean, whole	-	3.52	3.32
Wheat middlings	-	4.58	4.32
Hydrogenated fat	0.78	0.55	0.52
Minerals and vitamin supplements ^1^	1.63	1.76	1.69
Nutrient composition			
UFL (U/kg DM)	0.97	0.98	0.98
ENl (Mcal/kg DM)	1.55	1.59	1.53
Crude protein (% DM)	16.50	15.97	16.34
Metabolizable protein (% DM)	10.50	10.79	10.92
NSC (% DM)	36.80	44.07	43.06
NDF (% DM)	35.80	32.18	32.91

^1^ Herd A: during the dry period dairy cows received 70 g/d of a supplement contained 42.9% Ca_2_PO_4_; 28.6% urea; 14.3% MgO; 7.1% NaCl; 7.1% mineral vitamin supplement composited to provide 100,000 UI of vitamin A, 10,000 IU of vitamin D, 500 mg of vitamin E, 100 mg of Mn, 300 mg of Zn, 50 mg of Cu, 5 mg of I, 1 mg of Co, 3 mg of Se. During the lactation dairy cows received 300 g/d of a supplement contained, 27.5% NaHCO_3_, 20% CaCO_3_, 20% CaHPO_4_; 7% MgO; 2% NaCl; 10% mineral vitamin supplement composited to provide, 150,000 UI of vitamin A, 15,000 IU of vitamin D, 1000 mg of vitamin E, 100 mg of vitamin K, 100 mg of vitamin H1 50 mg of vitamin B1, 0.5 mg of vitamin B12, 500 mg of vitamin PP, 4000 mg of choline, 700 mg of Mn, 1200 mg of Zn, 200 mg of Cu, 20 mg of I, 2 mg of Co, 4 mg Se. Herd B: during the lactation dairy cows received a supplement composited to provide 132,000 UI of vitamin A, 33,000 IU of vitamin D, 650 mg of vitamin E, 770 mg of Mn, 1100 mg of Zn, 180 mg of Cu, 22 mg of I, 2.5 mg of Co, 3.6 mg of Se. During the dry period dairy cows received a supplement composited to provide 210,000 UI of vitamin A, 54,000 IU of vitamin D, 1000 mg of vitamin E, 1200 mg of Mn, 600 mg of Zn, 300 mg of Cu, 35 mg of I, 4.2 mg of Co, 5.5 mg of Se.

**Table 2 animals-12-01202-t002:** Extracted factors from factor analysis and proposed factor names.

Item ^1^	F1	F2	F3	F4	F5	F6	F7	F8
	De Novo/ Preformed	BCFA ^2^	LC-PUFA ^3^	alt RBH ALA ^4^	Production	SCFA ^5^	OCFA ^6^	RBH ^7^
Milk yield, kg	0.12	−0.21	−0.07	−0.12	**0.92**	−0.04	0.03	0.06
Milk composition, kg								
Fat	0.01	−0.12	−0.11	−0.04	**0.82**	−0.03	−0.03	−0.12
Protein	0.26	−0.28	−0.02	−0.04	**0.82**	−0.04	0.08	0.09
Individual fatty acids, g/100 g of total FAs								
SFA								
4:0	−0.19	0.26	−0.13	0.00	0.00	**0.65**	−0.22	−0.11
6:0	0.31	0.30	−0.21	0.12	−0.04	**0.75**	−0.07	−0.16
8:0	**0.62**	0.27	−0.12	0.20	−0.01	**0.65**	0.03	−0.12
10:0	**0.83**	0.17	0.02	0.17	0.03	0.37	0.13	−0.08
11:0	0.54	−0.30	0.08	0.14	−0.01	0.03	**0.71**	0.06
12:0	**0.89**	0.12	0.04	0.12	0.03	0.18	0.19	−0.06
13:0 iso	0.09	**0.67**	−0.03	0.11	−0.11	−0.04	−0.17	−0.27
13:0	**0.68**	−0.22	0.14	0.07	0.04	−0.13	**0.60**	0.06
14:0 iso	0.20	**0.75**	0.19	0.02	−0.14	0.16	−0.12	−0.24
14:0	**0.81**	0.12	−0.25	0.03	0.11	0.06	0.09	−0.09
15:0 iso	−0.09	**0.88**	−0.03	0.13	−0.10	0.07	−0.07	−0.14
15:0 ante	0.03	**0.82**	0.03	0.05	−0.17	0.08	0.03	0.06
15:0	0.49	−0.19	−0.05	0.09	0.04	−0.17	**0.76**	0.08
16:0 iso	−0.19	**0.64**	0.09	−0.09	−0.17	0.26	−0.32	−0.09
17:0 iso	**−0.66**	0.45	−0.02	0.00	−0.04	0.11	−0.07	0.18
17:0 ante	**−0.68**	0.36	0.03	−0.27	−0.07	0.28	−0.11	0.28
17:0	**−0.71**	−0.07	−0.22	−0.14	−0.08	0.27	0.37	0.16
18:0	**−0.61**	0.31	−0.28	0.43	−0.19	0.16	−0.22	0.01
MUFA								
10:1 (c9)	**0.85**	0.13	0.19	0.00	0.07	−0.02	0.01	−0.17
14:1 (c9)	**0.63**	−0.18	0.33	−0.24	0.20	−0.37	0.03	−0.07
16:1 (t9)	**−0.73**	0.00	0.32	−0.25	0.01	0.07	−0.04	0.17
18:1 (t6+t8)	−0.31	−0.03	0.14	0.08	0.01	−0.05	−0.03	**0.64**
18:1 (t9)	−0.15	−0.06	0.19	−0.05	−0.01	−0.11	0.01	**0.88**
18:1 (t10)	−0.11	−0.33	0.16	−0.03	0.03	−0.01	0.19	**0.65**
18:1 (t11)	−0.32	0.05	0.19	−0.03	−0.01	0.03	0.02	**0.80**
18:1 (c9)	**−0.76**	−0.17	0.41	−0.06	0.00	−0.18	−0.21	0.05
18:1 (c12)	0.16	−0.26	0.38	0.33	0.15	−0.04	0.11	**0.64**
18:1 (t16 + c14)	0.06	0.03	0.07	**0.68**	0.04	−0.14	0.07	0.42
PUFA								
18:2 (c9, c12)	−0.08	−0.22	**0.84**	−0.01	−0.04	−0.1	−0.09	0.21
18:3 (c9, c12, c15)	0.04	−0.07	**0.75**	0.07	−0.02	0.01	0.05	0.21
18:2 (c9, t11)	−0.07	0.07	**0.76**	−0.12	0.03	−0.22	−0.06	0.2
20:3 (c8, c11, c14)	0.19	0.06	**0.63**	0.20	0.07	−0.16	0.03	−0.01
20:4 (c5, c8, c11, c14)	−0.09	−0.10	**0.74**	0.11	−0.12	−0.01	0.09	0.1
Eigenvalue	8.89	5.10	4.92	2.35	2.76	2.66	2.31	3.98

^1^ c = cis; t = trans; SFA = saturated fatty acids; MUFA = monounsaturated fatty acids; PUFA = polyunsaturated fatty acids. ^2^ BCFA = branched chain fatty acids. ^3^ LC-PUFA = long chain—polyunsaturated fatty acids. ^4^ alt RBH ALA = alternative rumen biohydrogenation of α-linolenic acid. ^5^ SCFA = short chain fatty acids. ^6^ OCFA = odd chain fatty acids. ^7^ RBH = rumen biohydrogenation. Values > |0.60| are highlighted using bold.

**Table 3 animals-12-01202-t003:** Results from linear mixed model *(p*-values) for the extracted factors F1, F2, F3, and F4.

Traits ^1^	F1-De Novo/Preformed		F2-BCFA ^2^		F3-LC-PUFA ^3^		F4-alt RBH ALA ^4^	
	*p*-value	RMSE ^5^	*p*-Value	RMSE ^5^	*p*-Value	RMSE ^5^	*p*-Value	RMSE ^5^
Individual sources of variation ^6^								
DIM, classes ^7^	**<0.001**	0.931	0.069	1.244	0.098	0.655	0.126	0.675
Parity, classes ^8^	0.990		0.362		0.979		0.249	
Body measure								
BCS, classes	0.764	0.932	0.278	1.261	**0.007**	0.649	0.229	0.650
Ultrasound measurements								
pTAG, mg/g	0.199	0.926	**0.028**	1.252	0.120	0.661	**0.025**	0.667
PVA, mm^2^	0.106	0.906	0.336	1.261	0.128	0.663	0.997	0.681
PVD, mm	0.816	0.928	0.874	1.283	0.418	0.668	0.828	0.684
LD, mm	0.809	0.941	0.668	1.302	0.314	0.663	0.698	0.680
Hematochemical parameters								
Hematocrit, L/L	0.972	0.927	0.811	1.308	0.050	0.679	0.346	0.666
Energy-related metabolites								
Glucose, mmol/L	0.926	0.924	0.538	1.303	0.204	0.687	**0.019**	0.649
Cholesterol, mmol/L	0.398	0.922	0.046	1.292	0.504	0.687	0.667	0.667
NEFA, mmol/L	0.109	0.914	0.928	1.311	0.473	0.688	**0.015**	0.660
BHB, mmol/L	0.597	0.927	**0.46**	1.308	0.373	0.689	0.233	0.668
Urea, mmol/L	0.179	0.917	0.073	1.286	0.441	0.690	0.793	0.670
Creatinine, µmol/L	0.765	0.924	0.071	1.286	0.770	0.693	0.117	0.659
Liver function/hepatic damage								
AST, U/L	0.883	0.925	0.973	1.311	0.720	0.691	0.989	0.670
GGT, U/L	0.801	0.924	0.532	1.306	0.130	0.678	0.573	0.666
BILt, µmol/L	0.080	0.914	0.280	1.299	0.584	0.691	0.053	0.660
Albumin, g/L	0.099	0.910	0.360	1.302	0.627	0.690	0.119	0.654
ALP, U/L	0.337	0.919	0.267	1.297	**0.015**	0.674	**<0.001**	0.618
PON, U/mL	0.942	0.926	0.824	1.309	0.920	0.693	0.776	0.669
Oxidative stress metabolites								
ROMt, mgH_2_O_2_/100 mL	0.243	0.924	0.126	1.291	0.052	0.673	0.597	0.668
AOPP, µmol/L	**0.014**	0.896	**0.017**	1.266	0.177	0.677	0.080	0.659
FRAP, µmol/L	0.642	0.922	0.178	1.292	0.172	0.685	0.498	0.668
SHp, µmol/L	0.241	0.914	0.459	1.306	**0.044**	0.683	0.663	0.667
Inflammation/innate immunity								
Ceruloplasmin, µmol/L	**0.010**	0.913	**0.030**	1.280	**0.012**	0.664	0.611	0.668
PROTt, g/L	0.578	0.923	0.785	1.308	0.464	0.688	0.481	0.666
Globulins, g/L	0.748	0.926	0.361	1.300	0.962	0.693	0.354	0.662
Haptoglobin, g/L	0.873	0.925	0.338	1.296	0.060	0.680	0.626	0.667
MPO, U/L	0.320	0.916	0.126	1.284	0.895	0.693	0.498	0.667
Minerals								
Ca, mmol/L	0.432	0.919	0.702	1.307	0.785	0.691	0.381	0.665
P, mmol/L	0.122	0.910	0.333	1.305	0.690	0.692	0.844	0.668
Mg, mmol/L	0.302	0.928	0.361	1.305	0.450	0.690	0.541	0.668
Na, mmol/L	0.872	0.925	0.473	1.302	0.954	0.693	0.782	0.669
K, mmol/L	0.149	0.923	0.271	1.303	0.482	0.690	0.663	0.668
Cl, mmol/L	0.871	0.925	0.618	1.313	0.168	0.684	0.131	0.661
Zn, µmol/L	0.396	0.922	0.109	1.288	0.430	0.689	0.206	0.668

^1^ All explanatory variables are discretized according to the 25th, and 75th percentiles, therefore 3 levels and 2 df are absorbed by each linear model. BCS = body condition score (expressed as class); pTAG = predicted liver triacylglycerol; PVA = portal vein area; PVD = portal vein depth; LD = liver depth; NEFA = non-esterified fatty acids; BHB = β-Hydroxybutyrate; AST = aspartate aminotransferase; GGT = γ-glutamyl transferase; BILt = total bilirubin; ALP = alkaline phosphatase; PON = paraoxonase; ROMt = total reactive oxygen metabolites; AOPP = advanced oxidation protein products; FRAP = ferric reducing antioxidant power; SHp = total thiol groups; PROTt = total proteins; MPO = myeloperoxidase. ^2^ BCFA = branched chain fatty acids. ^3^ LC-PUFA = long chain—polyunsaturated fatty acids. ^4^ Alt RBH ALA = alternative rumen biohydrogenation of α-linolenic acid. ^5^ RMSE = root mean square error. ^6^ Model which considered only DIM, parity and herd/date as sources of variations. ^7^ DIM is constituted by 4 classes: class 1 ≤ 30; 30 < class 2 ≤ 60; 60 < class 3 ≤ 90; 90 < class 4 ≤ 120. ^8^ Parity is constituted by two classes: primiparous and multiparous. *p*-values < 0.05 are highlighted using bold.

**Table 4 animals-12-01202-t004:** Results from linear mixed model (*p*-values) for the extracted factors F5, F6, F7, and F8.

Traits ^1^	F5-Production		F6-SCFA ^2^		F7-OCFA ^3^		F8-RBH ^4^	
	*p*-Value	RMSE ^5^	*p*-Value	RMSE ^5^	*p*-Value	RMSE ^2^	*p*-Value	RMSE ^5^
Individual sources of variation ^6^								
DIM, classes ^7^	0.139	1.868	0.381	2.900	0.569	1.159	0.701	0.922
Parity, classes ^8^	0.013		0.689		0.839		0.468	
Body measure								
BCS, classes	0.850	1.783	0.744	2.848	0.195	1.162	0.057	0.920
Ultrasound measurements								
pTAG, mg/g	**0.030**	1.834	**0.044**	2.829	0.344	1.159	0.143	0.919
PVA, mm^2^	0.783	1.878	0.151	2.855	0.052	1.164	0.126	0.939
PVD, mm	0.665	1.880	0.350	2.890	0.227	1.167	0.738	0.938
LD, mm	0.929	1.918	0.895	2.938	0.815	1.166	0.521	0.949
Hematochemical parameters								
Hematocrit, L/L	0.569	1.978	0.386	2.954	0.220	1.184	0.695	0.955
Energy-related metabolites								
Glucose, mmol/L	0.670	1.957	0.678	2.938	**0.039**	1.159	**0.016**	0.921
Cholesterol, mmol/L	**0.004**	1.877	0.082	2.901	0.955	1.189	**0.016**	0.926
NEFA, mmol/L	0.146	1.908	0.403	2.924	0.352	1.177	0.899	0.954
BHB, mmol/L	0.457	1.970	0.613	2.956	**0.003**	1.135	0.253	0.948
Urea, mmol/L	0.777	1.966	0.700	2.954	0.108	1.167	0.673	0.953
Creatinine, µmol/L	0.103	1.950	0.292	2.941	0.886	1.189	0.384	0.947
Liver function/hepatic damage								
AST, U/L	0.754	1.976	0.972	2.965	0.678	1.185	0.689	0.951
GGT, U/L	0.480	1.961	0.177	2.913	**0.046**	1.162	0.145	0.944
BILt, µmol/L	0.576	1.970	0.148	2.928	0.155	1.168	0.853	0.950
Albumin, g/L	**0.037**	1.896	0.137	2.880	**<0.001**	1.066	**0.011**	0.909
ALP, U/L	0.863	1.964	0.798	2.962	0.105	1.171	0.232	0.945
PON, U/mL	0.168	1.953	0.379	2.944	0.187	1.177	0.752	0.951
Oxidative stress metabolites								
ROMt, mgH_2_O_2_/100 mL	0.724	1.965	0.455	2.950	0.527	1.177	0.387	0.948
AOPP, µmol/L	0.170	1.904	0.262	2.899	0.367	1.178	**0.015**	0.955
FRAP, µmol/L	0.618	1.934	0.597	2.909	0.991	1.189	0.912	0.957
SHp, µmol/L	0.378	1.958	0.400	2.947	**0.032**	1.159	0.504	0.944
Inflammation/innate immunity								
Ceruloplasmin, µmol/L	0.677	1.962	0.686	2.958	**0.016**	1.147	0.526	0.950
PROTt, g/L	0.705	1.968	0.800	2.962	**0.023**	1.157	0.395	0.949
Globulins, g/L	0.396	1.966	0.276	2.940	**0.021**	1.159	0.727	0.948
Haptoglobin, g/L	0.082	1.937	0.171	2.931	0.647	1.184	0.675	0.953
MPO, U/L	0.411	1.968	0.722	2.960	0.898	1.189	0.836	0.952
Minerals								
Ca, mmol/L	0.537	1.953	0.771	2.959	0.648	1.188	0.882	0.952
P, mmol/L	0.748	1.976	0.347	2.945	0.902	1.190	0.592	0.952
Mg, mmol/L	0.786	1.966	0.717	2.959	0.111	1.161	0.645	0.948
Na, mmol/L	0.596	1.957	0.654	2.955	0.510	1.184	0.143	0.947
K, mmol/L	0.263	1.962	0.072	2.914	0.742	1.187	0.495	0.949
Cl, mmol/L	0.441	1.969	0.171	2.931	0.915	1.188	0.765	0.951
Zn, µmol/L	0.737	1.971	0.171	2.931	0.972	1.189	0.090	0.949

^1^ All explanatory variables are discretized according the 25th, and 75th percentiles, therefore 3 levels and 2 degrees of freedom are absorbed by each linear model; BCS = body condition score (expressed as class); pTAG = predicted liver triacylglycerol; PVA = portal vein area; PVD = portal vein depth; LD = liver depth; NEFA = non-esterified fatty acids; BHB = β-Hydroxybutyrate; AST = aspartate aminotransferase; GGT = γ-glutamyl transferase; BILt = total bilirubin; ALP = alkaline phosphatase; PON = paraoxonase; ROMt = total reactive oxygen metabolites; AOPP = advanced oxidation protein products; FRAP = ferric reducing antioxidant power; SHp = total thiol groups; PROTt = total proteins; MPO = myeloperoxidase. ^2^ SCFA = short chain fatty acids. ^3^ OCFA = odd chain fatty acids. ^4^ RBH = rumen biohydrogenation. ^5^ RMSE = root mean square error. ^6^ Model which considered only DIM, parity and herd/date as sources of variation. ^7^ DIM is constituted by 4 classes: class 1 ≤ 30; 30 < class 2 ≤ 60; 60 < class 3 ≤ 90; 90 < class 4 ≤ 120. ^8^ Parity is constituted by two classes: primiparous and multiparous. *p*-values < 0.05 are highlighted using bold.

**Table 5 animals-12-01202-t005:** Summary of significant associations between milk fatty acid (FA) extracted factors and indicators of metabolic stress with brief physiological meaning description.

FA Factors (F) ^1^	Indicators of Metabolic Stress ^2^	Physiological Meaning
F1-de novo/ preformed	AOPP, µmol/L (↓) CP, µmol/L (↓)	Enzymatic activities decreased during inflammatory and oxidative stress
F2-BCFA	AOPP, µmol/L (↑) CP, µmol/L (↑) pTAG, mg/g (↓) Cholesterol, µmol/L (↑)	Negatively related with negative energetic balance and reduced feed intake Involvement in oxidative pathways Reported proinflammatory activities
F3-LC-PUFA	BCS, classes (↑) CP, µmol/L (↑)	Mirror of energy balance and lipid accumulation in the organism Positively associated with marker of inflammation and hepatic damage
F4-alternative RBH ALA	pTAG, mg/g (↓) NEFA, mmol/L (↓) Glucose, µmol/L (↑) ALP, U/L (↑)	Positively associated with marker of inflammation and hepatic damage Inverse relationship with negative energy balance and hepatic lipid accumulation
F5-production	pTAG, mg/g (↓) Cholesterol, mmol/L (↑) Albumin, U/L (↑)	Inverse relationship with negative energy balance and hepatic lipid accumulation
F6-SCFA	pTAG, mg/g (↓)	Dependent from glucose metabolism Inverse relationship with negative energy balance and hepatic lipid accumulation
F7-OCFA	GGT, U/L (↑) SHp, µmol/L (↑) Albumin, U/L (↑) BHB, mmol/L (↓) CP, µmol/L (↓) Globulins, U/L (↓)	Positively associated with antioxidant and anti-inflammatory activity Increased OCFA levels during hepatic damage Negatively associated to ketonemia, might be used as indicator of negative energy balance
F8-RBH	Glucose, µmol/L (↑) Cholesterol, mmol/L (↑) Albumin, U/L (↑) AOPP, µmol/L (↑)	Negatively related with negative energetic balance and reduced feed intake that reduced rumen enzymatic activities Enhanced by glucose availability Involvement in oxidative pathways

^1.^ BCFA = branched chain fatty acids; LC-PUFA = long chain—polyunsaturated fatty acids; RBH ALA = alternative rumen biohydrogenation of α-linolenic acid; SCFA = short chain fatty acids; OCFA = odd chain fatty acids; RBH = rumen biohydrogenation. ^2.^↑ = positively associated; ↓ = negatively associated; BCS = body condition score; pTAG = predicted liver triacylglycerol; NEFA = non-esterified fatty acids; BHB = β-Hydroxybutyrate; GGT = γ-glutamyl transferase; ALP = alkaline phosphatase; AOPP = advanced oxidation protein products; SHp = total thiol groups; CP = ceruloplasmin.

## Data Availability

All the dataset used in this study are available from the authors upon reasonable request.

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
