# Peer review of "Associations between Milk Fatty Acid Profile and Body Condition Score, Ultrasound Hepatic Measurements and Blood Metabolites in Holstein Cows"

_animals, 2022, doi:10.3390/ani12091202_

Round 1

Reviewer 1 Report

The main question addressed by the proposed manuscript is the problem of 'Metabolic disorders' in lactating dairy cows (Associations between milk fatty acids, body condition score, ultrasound hepatic measurements and blood metabolites in dairy Holstein cows).
The topic is relevant to the field. 'Metabolic disorders' are the main problem in high milk production. We should always think about how to prevent them. Compared to other publications, the proposed manuscript provides additional insight into the research questions. I have no idea for improvements regarding methodology. The conclusions are consistent with the presented arguments, and they address the main question posed. The references are appropriate. Line 397, delete a dot 'high plasma levels. (Figure 4h, p = 0.023).'

Line 467, add a dot '.' at the end of the sentence.

The tables and figures are correct and informative. I have no additional comments.

I have no additional suggestions for manuscript improvement

Author Response

The main question addressed by the proposed manuscript is the problem of 'Metabolic disorders' in lactating dairy cows (Associations between milk fatty acids, body condition score, ultrasound hepatic measurements and blood metabolites in dairy Holstein cows).

The topic is relevant to the field. 'Metabolic disorders' are the main problem in high milk production. We should always think about how to prevent them. Compared to other publications, the proposed manuscript provides additional insight into the research questions. I have no idea for improvements regarding methodology. The conclusions are consistent with the presented arguments, and they address the main question posed. The references are appropriate. Line 397, delete a dot 'high plasma levels. (Figure 4h, p = 0.023).'

Line 467, add a dot '.' at the end of the sentence.

The tables and figures are correct and informative. I have no additional comments.

I have no additional suggestions for manuscript improvement.

Authors. Thank you for the consideration given to our work. We made the revision you required.

Reviewer 2 Report

Dear Authors,

Your study is very interesting and offers new perspectives on the potential use of associations between milk FAs, US measurements, and blood metabolites as indicators of variation in energy and nutritional metabolisms in early lactating dairy cows.

However, I point out some aspects below that should be explained in more detail in material and methods, and discussions:

Material and methods

- It would be appropriate to describe the numerical distribution of lactating cows in the 4 DIM classes (class1≤30; .... 90 <class 4 <120) used in the mixed model analysis

- How many primiparous and how many pluriparous? (per herd and DIM classes)

- Please explain why in herd B (Table 1) two lactation diets are given: for primiparous and pluriparous? for two different genetic types? Given this consideration, how many herds did you include in the random effect of the model?

- Were the US measurements and blood sampling done on the same day that the milk samples were taken? Please specify.

Discussion

You should first discuss, in a first section and in a few essential points, your results with regard to the interpretation of metabolic alterations/variations in cows in early lactation, showing the advantages of their use in screening at farm level, already in this chapter, but their description, divided into numerous sub-chapters, does not exactly facilitate an overview when reading and understanding them.

You should also briefly point out the main implications of your study for the future in the conclusions.

Best Regards

Line

19                   components, the fatty

40                   , and

62                   a subsequent

78                   , therefore,

79                   status,

159                 were   was

186                 , and

230                 , and

242                 In order to To

354                 F4 and are

374                 (SCFA) and (?). Specifically

397                 , and

429                 , and

482                 , and

502                 , and

519                 , and               

592                 , therefore,

603                 the conversion

604                 the production

613                 , and

628                 , and

Reviewer 3 Report

Additional remarks:

The quality and the scientific significance of this manuscript are good. The text is comprehensive, but I find some problematic parts.

line 73: suggestion: please add the location of absorption (from gut) to this sentence!

lines 100-102: is this part really necessary?

Table 1: what the differences between “Lactation” and “Lactation” in Herd B? Are you measured the dry matter intake? Why interesting the diet composition in the dry period? What was differences between two farms in milk production? In this investigation measured 297 cows (5-120 DIM), how distributed the cows between in two herds? DIM was balanced in these farms? Did measure the crude fat content and fatty acid profile of fed TMR?

lines 127-130: how selected the cows (n=297) from the two herds? Parity?

lines 131-132: how measured the milk compositions? Percent (%) or daily kg presented fat and protein content (Supplementary Table 1), later it was used fat, protein daily yield (kg)!

 line 160: “g” instead of “G”

lines 165-171: no information of methods!!!

lines 175-179: this part belongs to statistical analysis section!

lines 211-227: is this part really necessary?

Table 2: please add correlation phrase in the title!

Table 3: the 1st RMSE index is wrong (2)!

Figures: how calculated the ranges? Please add more info in to the material and methods section!! How calculated the trendline? are you used only three points or all points? Presently, it seems, used only 3 points!

line 374: not finished the sentence (“… fatty acids (SCFA) and. Specifically,…”)!

line 422, 435 and other lines: de novo, cis trans: please put in italic style!

Author Response

Reviewer #3

Additional remarks:

The quality and the scientific significance of this manuscript are good. The text is comprehensive, but I find some problematic parts.

Authors. Thank you for the consideration given to our work.

 line 73: suggestion: please add the location of absorption (from gut) to this sentence!

Authors. Thank you, done.

lines 100-102: is this part really necessary?

Authors. Thank you for the suggestion. Based on concerns raised by reviewer #4 who asked for more details about the project, sampling procedures and sample size calculation, we prefer to keep this initial part.

Table 1: what the differences between “Lactation” and “Lactation” in Herd B? Are you measured the dry matter intake? Why interesting the diet composition in the dry period? What was differences between two farms in milk production? In this investigation measured 297 cows (5-120 DIM), how distributed the cows between in two herds? DIM was balanced in these farms? Did measure the crude fat content and fatty acid profile of fed TMR?

Authors. Thank you for your comments. The two “lactation” in Table 1 were related to the different diets for primiparous and multiparous. We apologize for the incomplete information that is now corrected. We deleted the columns related to dry period, that are non-informative in this work. The production of the two herds was similar (herd A: 36.89 ± 8.33 and herd B: 38.16 ± 7.29 kg/d). This information together with those related to the frequency’s distributions of DIM classes as well as parity order were included in the text (lines 130-131 and 254-256). We did not measure the crude fat content and FA profile of the fed TMR.

lines 127-130: how selected the cows (n=297) from the two herds? Parity?

Authors. Thank you for your comment. Cows were selected for DIM (<120) in the two herds, anyway, the parity order was balanced. Details on frequencies distribution for all individual sources of variation are now reported at lines 254-256.

lines 131-132: how measured the milk compositions? Percent (%) or daily kg presented fat and protein content (Supplementary Table 1), later it was used fat, protein daily yield (kg)!

Authors. Thank you. We corrected the Supplementary Table 1 with the correct unit of measure (kg).

 line 160: “g” instead of “G”

Authors. Thank you, done.

lines 165-171: no information of methods!!!

Authors. Thank you. We added the information you required in lines 165-168.

lines 175-179: this part belongs to statistical analysis section!

Authors. Thank you for the suggestion. We prefer to keep this part in a specific section dedicated to BCS assessment to avoid confusion with the mixed model used for the association analyses. Please consider that more information is provided by a recent paper – now cited in the text – recently accepted in JDS and going to be published soon (see line 180).

lines 211-227: is this part really necessary?

Authors. Thank you for the suggestion. We deleted the sentence.

Table 2: please add correlation phrase in the title!

Authors. Thank you. We rephrased the caption adding “correlation”.

Table 3: the 1st RMSE index is wrong (2)!

Authors. Thank you, done.

Figures: how calculated the ranges? Please add more info in to the material and methods section!! How calculated the trendline? are you used only three points or all points? Presently, it seems, used only 3 points!

Authors. Thank you for your comments. Information about the discretization in classes is given in lines 266-268. Regarding trendlines, they have been depicted according to the polynomial contrast carried out using the lme4 R package (p < 0.05) as detailed in lines 273-276. Having three levels, we tested linear and quadratic tendencies. This specification has been incorporated in all figures’ captions.

line 374: not finished the sentence (“… fatty acids (SCFA) and. Specifically,…”)!

Authors. Thank you, we corrected the sentence.

line 422, 435 and other lines: de novo, cis trans: please put in italic style!

Authors. Thank you for the suggestion. Anyhow, the Animals journal guidelines define that Latin words in the text do not need italic style.

Reviewer 4 Report

Dear Authors,

The present manuscript is well presented. 

My objection lies in the incomplete presentation of the animal selection criteria.

Is the sample size sufficient for a robust conclusion?

Did all animals for each farm participate in the study? 

if not which animals were selected and based on which criteria?

Author Response

Reviewer #4

Dear Authors,

The present manuscript is well presented. 

Authors. Thank you for the consideration given to our work. We made the revision according to the concerns raised in the PDF you attached.

My objection lies in the incomplete presentation of the animal selection criteria. Is the sample size sufficient for a robust conclusion? Did all animals for each farm participate in the study? if not which animals were selected and based on which criteria?

Authors. Thank you for your comments. Inclusion criteria are specified in lines 129-130 and 152-154: the total number of cows of the two farms was around 1000 (line 100), but in this study were recruited 297 of them on the basis of DIM (< 120) and US imaging (animals with liver abscesses or neoplastic masses were excluded. Also, animals with clinical diseases were not included. Concerning the sample size, this trial is part of a broader project that was approved by the Ethical committee of the ‘OPBA- Organismo Preposto al Benessere degli Animali of the Università Cattolica del Sacro Cuore’ and by the Italian Ministry of Health and a sample size calculation is mandatory to receive the approval (please see lines 100-110). Specifically, considering the variability of investigated traits as well as the experimental design based on a factorial design, with this sample size we achieve a margin of error < 5% with a CI of 95%.

Round 2

Reviewer 2 Report

Dear Authors,

I think that overall the presentation of MS has been well improved. I appreciate the efforts made to clarify some aspects of the study design and the usefulness of screening metabolic changes in cows in early lactation.

Best regards

Line

73                   They are the result of the mobilization

References    remove the point at the end of DOI